# The Role of Filial Piety in the Relationships between Work Stress, Job Satisfaction, and Turnover Intention: A Moderated Mediation Model

**DOI:** 10.3390/ijerph18020714

**Published:** 2021-01-15

**Authors:** Jianfeng Li, Hongping Liu, Beatrice van der Heijden, Zhiwen Guo

**Affiliations:** 1Business School, Hubei University, 368 Youyi Ave., Wuchang District, Wuhan 430062, China; b.vanderheijden@fm.ru.nl (B.v.d.H.); guozhiwen@hubu.edu.cn (Z.G.); 2School of Economic and Business Administration, Central China Normal University, 152 Luoyu Ave., Hongshan District, Wuhan 430079, China; liuhongping@mail.ccnu.edu.cn; 3Institute for Management Research, Radboud University, 6525 AJ Nijmegen, The Netherlands; 4School of Management, Open University of the Netherlands, 6419 AT Heerlen, The Netherlands; 5Faculty of Economics and Business Administration, Ghent University, 9000 Ghent, Belgium; 6Kingston Business School, Kingston University, London KT11LQ, UK

**Keywords:** filial piety, reciprocal filial piety, authoritarian filial piety, work stress, job satisfaction, turnover intention

## Abstract

In China, filial piety, which usually refers to showing respect and obedience to parents, has exerted an important effect in the relationship between work stress and turnover intention. However, the mechanism behind this effect is still unclear. To address this gap in the existing literature, we developed and tested a moderated mediation model of the relationship that work stress shares with job satisfaction and turnover intention. In accordance with the dual filial piety model and the stress-moderation model, our hypothesized model predicted that the mediating effect of job satisfaction on the relationship between work stress and turnover intention would be moderated by reciprocal filial piety (RFP) and authoritarian filial piety (AFP). The analytic results of data that were obtained from 506 employees of manufacturing industries in China supported this model. Specifically, RFP and AFP, as a contextualized personality construct, positively moderated the direct relationship between work stress and turnover intention as well as the corresponding indirect effect through job satisfaction. In particular, RFP and AFP strengthened the positive effect of work stress on turnover intention. Based on these findings, recommendations to help employees fulfill their filial duties and reduce the effect of work stress on turnover intention among employees of Chinese manufacturing industries are delineated.

## 1. Introduction

In China, there have been many recent reports of a filial son who killed his sick mother as a result of high levels of work stress [1] and another son who resigned from his job to care for his sick parents at home [2]. Therefore, some employers have provided employees with optimal conditions and accessible resources to ease their caregiving burden, enhance job satisfaction, and retain them in the organizational workforce [3]. Thus, filial piety may be inextricably linked to work stress, job satisfaction, and turnover intention. However, the role that filial piety plays in individual attitudes and organizational behaviors is still unclear [4], and we are not aware of any prior empirical research that has specifically focused on the relationships that filial piety shares with work stress, job satisfaction, and turnover intention.

In contemporary China, a filial son is typically used to refer to a son who does his best to support his parents and comply with their wishes [5], and obedience and loyalty are fundamental characteristics of a filial person [6]. Since filial piety is the root of all virtues and foundation of familial and social order, it has emerged as the most important indicator of good deeds and moral conduct in Chinese society [7]. Filial piety has deeply impacted individual attitudes and organizational behaviors. In Chinese society, “seeking loyalists who are filial sons” has emerged as an important criterion that managers and rulers use to select highly talented persons (or those with great potential). In particular, filial piety has emerged as the basic moral principle according to which Huawei, Lenovo, and Cheung Kong treat their employees [8]. Further, the issue of undergoing a compulsory course on filial piety was the subject of much attention during the 2018 Civil Service Examination in China. In the eyes of many Chinese people, a person who does not demonstrate filial obedience to his parents is also unlikely to be loyal to his superiors and bosses.

Filial piety is a complex construct that has multiple implications for fields such as society and politics, ethics, and religion [9]. From a social psychological perspective, Bedford and Yeh, in the dual filial piety model (DFPM), argued that filial piety is a contextualized personality construct and essentially consists of two fundamental components: reciprocal filial piety (RFP) and authoritarian filial piety (AFP) [10]. The DFPM is considered to be the model that best delineates the theoretical reasoning that underlies filial piety [11]. The historical and cultural traits of RFP and AFP have been preserved through intergenerational transmission in China [12], and the construct holds the most prominent place in Chinese ethics [13].

Past empirical research has shown that extreme work stress has a significant impact on physical and mental health [14,15] and is related to negative work-related attitudes such as poor job satisfaction and high turnover intention [15,16]. In addition, recent research has also shown that filial piety (i.e., RFP and AFP) is correlated with personal stress (e.g., depression and anxiety) [17], that personality moderates the relationships between work stress and strains (psychological, physiological, and behavioral outcomes) [14], and that job satisfaction is a fundamental predictor of turnover intention [16,18]. Accordingly, this study aimed to empirically examine the relationships between the two types of filial piety (i.e., RF and AFP), work stress, job satisfaction, and turnover intention using a moderated mediation model (see Figure 1). We intended to contextualize the findings on the relationships between individual attitudes and organizational behaviors within the framework of the DFPM, which is an indigenous psychological theory.

## 2. Theoretical Background and Hypotheses

### 2.1. The Mediating Effect of Job Satisfaction on the Relationship between Work Stress and Turnover Intention

Turnover intention refers to the extent to which an employee intends to leave his/her employer [19], and it is a major predictor of actual turnover [20]. According to Price and Mueller’s causal model of turnover, work stress is an important negative determinant of job satisfaction [19]. Further, by causing job dissatisfaction, work stress may lead to job search behaviors, turnover intention, and eventually, actual turnover [21]. However, according to the unfolding model of voluntary employee turnover that has been proposed by Lee and Mitchell, job search behaviors are not necessarily an antecedent to turnover [22]. Similarly, Steers and Mowday’s turnover model consider job satisfaction to be an important negative determinant of turnover intention [23]. In addition, Danna and Griffin’s health and well-being model considers that work stress is an important positive predictor of job-related dissatisfaction, which in turn leads to more absenteeism [24].

Work stress is also an important predictor of turnover intention in Chinese societies. In particular, work stress has a direct effect on job satisfaction, which in turn affects turnover intention among nurses [25]. Recent empirical research has shown that job satisfaction partially mediates the relationship between job stressors and turnover intention among airport traffic controllers [26] as well as between job stressors and turnover intention among airport security screeners [27]. Similarly, job satisfaction partially mediated the relationship between work stress and turnover intention among nurses who provided long-term care [28]. On the other hand, in a study among rural health workers [29], job satisfaction totally mediated the relationship between work stress and turnover intention. Therefore, based on the theoretical outline given above, we formulated the following hypothesis:

**Hypothesis** **1** **(H1):**
*Work stress will be negatively related to job satisfaction, which in turn will (partially) mediate the relationship between work stress and turnover intention.*


### 2.2. RFP and AFP as Moderators in the Relationships between Work Stress, Job Satisfaction, and Turnover Intention

Personality refers to ‘the dynamic organization within the individual of those psychophysical systems that determine his unique adjustments to his environment’ (p. 48) [30], and it can be divided into two fundamental kinds: context-specific, or contextualized personality and general, or non-contextualized personality [31]. Heller et al. defined contextualized personality as ‘stable patterns of thought, feelings, and behaviors that occur repeatedly within a given context’ [32] (p. 1229). This definition reveals that personality is expressed differently across various social roles and contexts. Dunlop considered that personality appears from the interaction between the individuals and surroundings and manifests in different ways depending upon goals and motivations as well as traits [33]. As these goals or motivations reflect certain interpersonal relationship or social context, social roles are considered as an ideal agent to contextualize personal goals or motivations [10]. Furthermore, social roles encompass two domains: status and belongingness, and by status and belongingness motivations, one can achieve interpersonal relatedness and social belonging to meet basic psychological needs [34].

Based on above mentioned theoretical backgrounds, Bedford and Yeh viewed filial piety as a contextualized personality, and they suggested that filial piety (i.e., RFP and AFP) can contextualize these two fundamental human needs: interpersonal relatedness and social belonging by parent–child social roles [10]. In the dual filial piety model (DFPM), RFP is a kind of genuine parent–child affection that results from long-term positive parent–child interactions; and AFP entails hierarchical intergenerational relationships between family members who play different roles [10]. Bedford and Yeh argued that RFP entails egalitarian parent–child relationships, whereby both parties understand each other and meet the needs of interpersonal relatedness through interaction and communication; AFP entails a hierarchical parent–child relationship, whereby children learn to meet their needs for social belongingness and collective identity from their parents [10]. In other words, RFP reflects the individual motives to fulfill interpersonal relatedness based on parent–child interpersonal connections; AFP reflects individual motives to achieve social identity and collective belonging based on abiding by social norms and role ethics.

Personality has a lasting and profound influence on people’s attitudes, affections, and behaviors [30], and it can moderate the relationship between work stress and strains [35]. Past studies have shown that work stress comprises a discrepancy between perceptions and desires [36], and it occurs when one is confronted with more demands at work and/or at home than what one can cope with, which may lead to negative psychological outcomes [37]. Additionally, conducting filial piety, being a contextualized personality, often means to spend time and energy in accompanying and caring for parents. As a result, doing so may subtract from available resources (e.g., time and energy) for work demands, which may lead to higher work stress. Accordingly, we speculated that filial piety (i.e., RFP and AFP), as a contextualized personality construct, might moderate the relationship between work stress and strains.

More specifically, RFP is based on the Confucian principles of “repaying parents for their gracious act of bearing and rearing children (bao)” and “favoring the intimate.” These principles emphasize the repayment of intergenerational intimate affection and love, and they are interpreted to be a spontaneous revelation of human nature. RFP is a reflection of human instinct, which reveals the phenomenon that people pay a debt of gratitude for whoever is good to them. In Chinese societies, the principle of “bao” (reciprocity) is an important foundation of social relations, and people believe that there is a reciprocate action (e.g., love and hate, reward and punishment) and a definite causal relationship between human beings, and even between human beings and supernatural beings [38]. Based on the principle of “bao”, people believe that they should be obedient to and repay their parents because they have got so much care from them, especially in early childhood. By this reciprocate action, people learn to fulfill interpersonal relatedness from family to society. For the sake of repaying their parents and fulfilling this intimate interpersonal relatedness, people will spend much time and energy in satisfying their parents’ demands, especially in such time that their parents are old and weak and need more help and support from them.

AFP entails hierarchical intergenerational relationships between family members who play different roles. It is based on the Confucian principle of “respect for superiors,” and it reflects a type of social identification that is acquired by obeying authorities and complying with social norms. For Chinese people, the most primitive and important authority figures are their parents, and showing obedience to parents is instilled from their childhood on. In Chinese society, “respect for parents” is not only respected by people, but also protected by laws. It is stipulated, in General Principles of the Civil Law, Marriage Law, and Law of the People’s Republic of China on Protection of the Rights and Interests of the Elderly, that the elderly are mainly supported by their families, and that family members have duties and obligations to care for the elderly. Whoever you are, once you violate the principle of “respect for parents”, you are to be condemned and punished by the community, and cannot gain social identification. For achieving collective identity, people will comply with the principle of “respect for parents” and spend time and energy in meeting their parents’ demands when their parents need help and support from them. As a result, doing so may subtract from available resources (e.g., time, energy) for work demands, which will lead to higher work stress.

In practice, ‘adult children should often visit or greet their elderly parents’ is explicitly stated in the legal provisions of the Law of the People’s Republic of China on Protection of the Rights and Interests of the Elderly, which was enacted on 1 July 2013. Recent research shows that there are 54% of elderly people whose sources of income mainly come from their adult children [39], and that adult children with higher levels of RFP/AFP provide greater emotional and financial support to their elderly parents than non-filial ones (lower levels of RFP and AFP) [40]. Accordingly, those with high levels of RFP/AFP may spend more time and energy in conducting filial duties and obligations than non-filial ones in daily life. As a result, doing so may subtract from available resources (e.g., time, energy) for work demands, which will deepen role conflicts between being a competent employee and being a filial person (e.g., it requires to spend time and energy to accompany and care for sick parents). As a result, this may trigger high work stress and negative psychological outcomes (e.g., low job satisfaction and high turnover intention). Furthermore, filial piety contains the connotation of dutifulness and self-discipline, which the conscientiousness trait of Big Five Personality also encompasses. Indeed, past studies have shown that RFP/AFP is positively related to conscientiousness [41], and that the negative effect of perceived role conflict on job satisfaction was enhanced by conscientiousness [42].

Hence, based on these speculations, we formulated the following hypotheses, and the moderated mediation model that was empirically tested in this study is presented in Figure 2.

**Hypothesis** **2a** **(H2a):**
*The negative relationship between work stress and job satisfaction will be moderated by RFP. Specifically, this relationship will be stronger among individuals with higher rather than lower levels of RFP.*


**Hypothesis** **2b** **(H2b):**
*The negative relationship between work stress and job satisfaction will be moderated by AFP. Specifically, this relationship will be stronger among individuals with higher rather than lower levels of AFP.*


**Hypothesis** **3a** **(H3a):**
*The positive relationship between work stress and turnover intention will be moderated by RFP. Specifically, this relationship will be stronger among individuals with higher rather than lower levels of RFP.*


**Hypothesis** **3b** **(H3b):**
*The positive relationship between work stress and turnover intention will be moderated by AFP. Specifically, this relationship will be stronger among individuals with higher rather than lower levels of AFP.*


Since we hypothesized that job satisfaction would (partially) mediate the relationship between work stress and turnover intention (Hypothesis 1) and that filial piety (i.e., RFP, AFP) would moderate the relationship between work stress and job satisfaction (Hypothesis 2a,b), by extension, we also predicted that job satisfaction would offer a more viable explanation for the relationship between work stress and turnover intention as an individual’s filial piety increases. That is, in combination, the rationales behind Hypotheses 1 and 2a,b combine to support the moderated mediation model depicted in Figure 2. More specially, we proposed the following hypotheses:

**Hypothesis** **4a** **(H4a):**
*RFP will moderate the extent to which job satisfaction mediates the relationship between work stress and turnover intention: job satisfaction is more likely to mediate this relationship when RFP is higher than when RFP is lower.*


**Hypothesis** **4b** **(H4b):**
*AFP will moderate the extent to which job satisfaction mediates the relationship between work stress and turnover intention: job satisfaction is more likely to mediate this relationship when AFP is higher than when AFP is lower.*


## 3. Method

### 3.1. Sample and Procedure

This study was conducted across manufacturing firms in China because the average turnover rate has been higher in this industry than in other industries across the past few years. The sample consisted of 506 participants who were working in Chinese manufacturing industries. All the participants were Chinese citizens, and they were recruited using convenience sampling. All the questionnaires were in Chinese, and we collected data through two simultaneous means.

First, we collected data from the employees of three electronic manufacturing firms in Shenzhen. With the permission of the three firms’ top managers and help of their human resource managers, all the participants were informed about the purpose of the survey and assured about the confidentiality of their responses and their right to withdraw their consent to participate in the study. They were given time to voluntarily and anonymously complete the survey in their respective workplaces. The response rates across the three firms were 84.0% (126/150), 84.3% (118/140), and 85.7% (120/140). All the employees of these firms who participated in this study were residing in China at the time of the study. After 58 invalid questionnaires (i.e., those with numerous missing responses) were excluded, a final pool of 306 valid responses was analyzed.

Second, we trained 12 assistants to collect data from their acquaintances who were employed in manufacturing industries across the nation. They collected data from 210 acquaintances through emails. After 10 invalid questionnaires (i.e., those with numerous missing responses) were excluded, a pool of 200 valid responses was analyzed.

Chi-squared tests revealed that there was no significant difference between the two samples (*n* = 306, *n* = 200) with regard to gender (χ^2^ = 3.02, *p* > 0.05), age (χ^2^ = 44.47, *p* > 0.05), duration of tenure (month) (χ^2^ = 67.76, *p* > 0.05), educational level (χ^2^ = 10.19, *p* > 0.05), and marital status (χ^2^ = 0.58, *p* > 0.05). Therefore, the two samples were combined into a single sample for all further analyses. Further, 54.7% of the participants were men. The average age of the respondents was 28.1 years (*SD* = 6.2); the average duration of tenure with their current employer was 31.0 months (*SD* = 42.5). Additionally, 45.7% and 44.7% of the participants had received higher education and were married, respectively.

### 3.2. Measures

Filial piety. Filial piety was assessed using a 9-item (α = 0.70) Chinese adaptation and revision of Yeh and Bedford’s scale [13,41]. It consists of two dimensions: RFP and AFP. RFP subsumes four items (e.g., “Be grateful to parents for raising me”; α = 0.82), whereas AFP comprises five items (e.g., “One should give up personal interests to fulfill parental expectations”; α = 0.67). The response categories range from 1 (“strongly disagree”) to 7 (“strongly agree”).

Work stress. Work stress was assessed using a 6-item (α = 0.87) adaptation and revision of Parker and DeCotiis’ scale that measures time pressure and job anxiety (e.g., “My job gets to me more than it should”) [43]. The response categories range from 1 (“strongly disagree”) to 7 (“strongly agree”).

Job satisfaction. An adaption of Hackman and Oldham’s 3-item (α = 0.92) measure of global job satisfaction (e.g., “Generally speaking, I am very satisfied with my job”) [44] was used to assess job satisfaction. The response categories range from 1 (“strongly disagree”) to 7 (“strongly agree”).

Turnover intention. Kelloway et al.’s 4-item (α = 0.93) measure of turnover intention (e.g., “I am planning to look for a new job”) [45] was used to measure turnover intention. The response categories range from 1 (“strongly disagree”) to 7 (“strongly agree”).

### 3.3. Control Variables

In this study, age, gender (1: male, 0: female), education (1: lower educational level, 0: higher educational level), duration of tenure with current employer, and marital status (1: married, 2: unmarried, 3: divorced) served as control variables. More specifically, Van der Heijden et al. found that age and duration of tenure have a significant influence on occupational turnover intention [46]. Zhou and his associates found that men report lower levels of job satisfaction than women [47], and Cui found that demographic characteristics such as age, gender, education, duration of tenure, and marital status influence turnover intention [48].

In addition, previous research that has been conducted in China has revealed that (a) traditional values tend to be negatively related to socioeconomic status, (b) filial piety is deeply rooted in the values of the elite sections of society rather than in the folk values of mass culture [49], and (c) daughters demonstrate greater filial piety toward their older parents than sons [50]. Indeed, as a result of China’s one-child policy, daughters have begun to play the role of a primary caretaker of their parents in contemporary China.

### 3.4. Data Analysis

We followed Hair et al.’s recommendations [51] to examine the discriminant validity of our measurement model using structural equation modeling (SEM). The following model fit indices were computed: normed chi-square statistic (𝜒^2^/*df*), goodness-of-fit index (GFI), Tucker-Lewis index (TLI), comparative fit index (CFI), root mean square error of approximation (RMSEA), and standardized root mean square residual (SRMR). As a rule of thumb, 𝜒^2^/*df* values that are ≤3 [51], GFI, TLI, and CFI values that are >0.90 [52], and RMSEA [53] and SRMR values that are ≤0.08 [54] are indicative of a good model fit.

We conducted a series of confirmatory factor analyses to investigate whether all the variables that were examined in this study were distinct. When compared to other models, the proposed five-factor structure (i.e., RFP, AFP, work stress, job satisfaction, and turnover intention) was found to be a significantly better fit for the data, 𝜒^2^/*df* = 582/199 = 2.92, *p* < 0.001, GFI = 0.902, TLI = 0.924, CFI = 0.935, RMSEA = 0.062, SRMR = 0.049. This finding suggested that all the study variables were distinct from one another.

We followed Podsakoff et al.’s suggestions [55] to control the effects of common-method variance (CMV). First, to enhance the credibility of the measurements that were used in this study, we used well-validated assessments. Second, we allowed all the participants to anonymously complete the questionnaires. Furthermore, we conducted Harman’s single-factor test to examine CMV; it refers to a type of confirmatory factor analysis in which all the variables are specified to load onto one common factor. The one-factor model was a very poor fit for the data, 𝜒^2^/*df* = 3994/211 = 18.93, *p* < 0.001, GFI = 0.506, TLI = 0.296, CFI = 0.357, RMSEA = 0.188, SRMR = 0.183. This indicated that a majority of the variance in our model was not explained by one single factor.

## 4. Results

Means, standard deviations, and reliability and correlation coefficients for all the study variables are presented in Table 1. We used hierarchical multiple regression analysis to test Hypothesis 1 and hierarchical moderated regression analysis to test Hypotheses 2 (i.e., 2a and 2b) and 3 (i.e., 3a and 3b). In particular, we followed Preacher, Rucker, and Hayes’ recommendations [56] to test moderated mediation models. In order to compute interaction terms and reduce the likelihood of multi-collinearity, we followed Wen and Ye’s suggestions [57] to transform the main study variables (i.e., RFP, AFP, work stress, job satisfaction, and turnover intention) into standardized z-scores (i.e., ZRFP, ZAFP, ZWS, ZJS, and ZTI) and created two interaction terms: ZRFP × ZWS and ZAFP × ZWS (ZRFP, ZAFP, ZWS, ZJS, and ZTI are the standardized z-scores of RFP, AFP, work stress, job satisfaction, and turnover intention, respectively). All variance inflation factors were less than 1.3; this indicated that our data did not violate the assumptions of regression analysis that pertain to multi-collinearity.

Hypothesis 1 stated that work stress would be negatively related to job satisfaction, which in turn would mediate the relationship between work stress and turnover intention. In order to reduce the Type I error rate and to improve the power of the test, we followed Wen and Ye’s suggestions [58] to test this mediation effect. In particular, Step 1, Baron and Kenny’s method [59] was used to successively test the following coefficients: *c* (*H*_0_: *c* = 0), *a* (*H*_0_: *a* = 0), *b* (*H*_0_: *b* = 0), and *c’* (*H*_0_: *c’* = 0) among the respective regression equations (*Y* = *cX* + *e*_1_, *M* = *aX* + *e*_2_, and *Y* = *c’X* + *bM* + *e*_3_); if either *a* or *b* was insignificant, Step 2 was conducted for which the Bootstrap method was used to test *ab* (*H*_0_: *ab* = 0). The results of the hierarchical multiple regression analysis for testing this mediation model are presented in Table 2.

It can be inferred from Table 2 that work stress was significantly related to turnover intention (*β* = 0.32, *p* < 0.001), when age, gender, education, tenure, and marital status were controlled for. This suggested that our data met the first condition for mediation analysis. Furthermore, job satisfaction was significantly related to turnover intention (*β* = −0.57, *p* < 0.001), and work stress was significantly related to job satisfaction (*β* = −0.30, *p* < 0.001). Thus, our data also met the other two requirements for mediation analysis. Finally, when both work stress and job satisfaction were simultaneously entered into the model, both the main effects remained significant (*β* = 0.18, *p* < 0.001; *β* = −0.52, *p* < 0.001). This indicated that job satisfaction partially mediated the relationship between work stress and turnover intention. These outcomes fully supported Hypothesis 1.

Hypotheses 2 (i.e., 2a and 2b) and 3 (i.e., 3a and 3b) predicted that the relationships that work stress shares with job satisfaction and turnover intention would be moderated by RFP and AFP. We tested the corresponding moderated mediation model as per Preacher et al.’s suggestions [56]. First, we examined whether work stress is significantly related to job satisfaction and turnover intention. It can be inferred from Table 2 that these requirements were met. Second, we examined whether the interactions between work stress and each moderator has an effect on the relationship between job satisfaction and turnover intention. The results of moderated regression analysis, which was conducted to examine the interaction effect between work stress and each moderator (i.e., RFP and AFP) on the relationship between job satisfaction and turnover intention, are presented in Table 3. The results revealed that RFP and AFP did interact with work stress and significantly predicted job satisfaction (*β* = −0.19, *p* < 0.001; *β* = −0.15, *p* < 0.01). Therefore, these results supported Hypothesis 2a,b. Table 3 also shows that both the moderators, namely, RFP and AFP, interacted with work stress and significantly predicted turnover intention (*β* = 0.15, *p* < 0.01; *β* = 0.18, *p* < 0.01). Hence, these results supported Hypothesis 3a,b.

Preacher et al.’s third condition for moderated mediation analysis [56] had already been met as job satisfaction was negatively related to turnover intention (see Table 2). To further investigate a possible moderated mediation effect, we examined whether our data fulfilled the fourth condition. Specifically, we examined whether the magnitude of the conditional indirect effect of work stress on turnover intention through job satisfaction was different among those with high and low levels of RFP and AFP. In accordance with Preacher et al.’s suggestions [56], we defined high and low levels of RFP and AFP as scores that lay one standard deviation above and below the mean, respectively. Estimates, standard errors, z-scores, and significance values for the conditional indirect effects are presented in Table 4. It can be inferred from this table that the indirect effect of work stress on turnover intention through job satisfaction was significant albeit only among those with high levels of RFP (conditional indirect effect = 0.26, *p* < 0.01) and AFP (conditional indirect effect = 0.21, *p* < 0.01). Hence, these results supported Hypotheses 4a,b.

## 5. Discussion

In response to calls for more research studies that examine the role that filial piety plays in individual attitudes and organizational behaviors [4], we tested a moderated mediation model of the relationship between work stress and turnover intention. To the best of our knowledge, this study comprises the first empirical study to explain the influence of filial piety on individual attitudes from an indigenous psychological perspective. Therefore, this study contributes to the already existing literature in two ways. First, we found that work stress (i.e., time pressure and job anxiety) is negatively related to job satisfaction and positively related to turnover intention. This finding suggests that high levels of time pressure and job anxiety, which very commonly originate from role conflict (e.g., long working hours may cause one to spend less time with family members), often lead to high levels of turnover intention in contemporary Chinese society.

Past studies have measured job stress primarily in terms of work overload, job requirement, role conflict, role ambiguity, occupational climate, and working environment [25,26,27,28,29]. In contrast, our study focused on the elements of time pressure and job anxiety as these have emerged as primary sources of job stress in contemporary China. In particular, in China, the rate at which wage earners engage in overtime work was as high as 42.2% in 2017 [60]. Additionally, in 2018, the average duration of time that individuals spent with and cared for family members (e.g., children, parents) was only 53 min per day because they had extended working hours [61]. Approximately 80% of Chinese people experience moderate levels of anxiety. This is primarily attributable to the following factors: the need to support oneself and one’s parents, medical care, career development, and children’s education [62].

Past research has already shown that, among those with Chinese background, work stress has a direct impact on depressed mood and job satisfaction, which in turn impact turnover intention [25]. It has also been demonstrated that workplace stress negatively affects job satisfaction, which in turn increases turnover intention [29]. Accordingly, our empirical findings also contribute to the growing body of research and extend previous scholarly research on the influence of work stress on job satisfaction and turnover intention within the Chinese context.

Second, by examining the role of filial piety as a moderator of the relationship that work stress shares with job satisfaction and turnover intention, we have explicitly addressed Lin et al.’s concerns [4] about the role that filial piety plays in individual attitudes and organizational behaviors. In doing so, we have also extended previous scholarly work that focused on the moderating role of personality in the relationship between work stress and strains [35]. We used the DFPM [41] as the underlying framework of our study and found that RFP and AFP, as a contextualized personality construct, positively and partially moderated the direct effect of work stress on turnover intention and the corresponding indirect effect through job satisfaction. More specifically, RFP and AFP strengthened the positive effect of work stress on turnover intention.

Our findings comply with longstanding recommendations to include the moderating effects of personality in occupational stress-strains relationship [14,35]. In particular, we found support for the contention that personality can indeed function as moderators of the relationships between work stress and strains (e.g., job satisfaction, turnover intention). In this manner, the present study extended prior research on filial piety to a Chinese context.

Traditionally, the principle of “bao” is a foundation of Chinese social relations, and filial piety is the most appropriate reflection of it [38]. As Tang has noted, ‘filial piety, love and respect for one’s parents, is not biological but is moral, being based upon a sense of obligation, or a debt of gratitude’ [63]. Chinese people consider it their duty to repay and support their older parents. Thus, when older parents need their adult children’s help and support, adult children with high rather than low levels of filial piety may invest more time and energy in caring for their parents as an expression of their gratitude to them. In China, a filial person is respected, and a non-filial one is condemned. However, perceived time pressures and job anxiety may worsen if being a competent employee is hard to be reconciled with being a filial person. In such circumstances, high levels of filial piety can exacerbate the adverse effects of work stress on turnover intention.

Filial piety emphasizes not only mutual interdependence among family members, but also parents’ authority and the hierarchical structure of the family. Specifically, the measurement of RFP in the present study emphasized the mutual interdependence and equality of parent–child relationships that are rooted in intimacy and the drive to meet the psychological need for relatedness of both parents and children [10].

The following content is an excerpt from the classic Chinese literary work, entitled *Nosta**lgia*: ‘when I was young, nostalgia was a tiny stamp, me on this side, mother on the other side. However, later on, nostalgia was a low, low grave, me on the outside, mother on the inside…’ [64]. RFP is founded on the love, care, and attachment that children share with their parents. Children with high rather than low levels of RFP share a stronger emotional connection with their parents. Accordingly, people with high levels of filial piety may invest more time and energy in fulfilling parental demands, which in turn may affect job satisfaction and turnover intention.

In the present study, the measurement of AFP emphasized parental authority, children’s obedience to role obligations that are grounded in hierarchical familial structures, and the drive to meet the psychological needs for social belongingness and collective identity through the process of socialization [10]. Children with high rather than low levels of AFP are more conscious about social norms and ethics, and AFP is positively correlated with personal stress (e.g., depression and anxiety) [17]. Accordingly, in order to achieve social identity and collective belongingness that necessitates filial behaviors, people with high levels of AFP may suppress their own desires to comply with parental wishes, and they may invest more time and energy in meeting parental demands. This in turn may affect job satisfaction and turnover intention.

### 5.1. Limitations and Suggestions for Future Research

Although our findings serve as a useful baseline for further investigations in Chinese societies, the present study also has several limitations. First, the present research study used a convenience sample of employees who worked in only manufacturing enterprises; this might have led to sampling bias. In addition, similar to past studies, we used cross-sectional data to examine the mediating effect of job satisfaction on the relationship between work stress and turnover intention. However, the use of cross-sectional data to test mediation models may produce biased fit indices [65]. Therefore, future studies must use cross-lagged models to compute more accurate fit indices for mediation effects.

Second, China is undergoing a drastic social transition; the one-child policy that had been implemented for more than 30 years was terminated in 2016. Further, the rate of urbanization increased from 21.1% in 1982 to 59.6% in 2018, and the old-age dependency ratio increased from 8.0% in 1982 to 16.8% in 2018 [66]. The traditional structure of Chinese families has been substantially deconstructed and reconstructed; concordantly, the meaning of filial piety has evolved extensively in accordance with changing family structures. As a result of the urbanization and globalization of business, the number of Chinese nuclear families has been decreasing, and the number of single-person households has been increasing [67]. Therefore, some researchers have contended that the factor structure of Yeh and Bedford’s DFPM may need to be revised in accordance with rapid modernization and the prevalence of conflicting family policies in China [13]. It has also been contended that a new measure of filial piety should be developed because existing measures of this construct fail to capture the aforementioned important changes [68]. Thus, alternative measures of filial piety must be developed in future research studies.

### 5.2. Practical Implications

First, since China is a family-oriented society, employers in Chinese manufacturing industries must provide necessary organizational support (e.g., provide housing, reduce overtime work) to employees to help them fulfill their filial duties to their parents and family members. As reported by the Chinese media, many employees who were unable to provide adequate care to their sick parents and family members resigned from their jobs because they had no other alternative solution. As a result of high levels of work stress, 72.9% of Chinese young adults are unable to fulfill their wishes to support their older parents [69], which may reduce their psychological well-being.

Second, employers should pay more attention to key employees with high levels of filial piety in accordance with the traditional Chinese practice of “seeking loyalists who are dutiful persons”. At present, the State Pension System has not yet been completely established. Therefore, most old parents in contemporary China rely solely on the support of their adult children [40]. For this reason, the Law of the People’s Republic of China on Protection of the Rights and Interests of the Elderly, which was revised on 1 July 2013, requires adult children to regularly visit or greet their parents. Therefore, employers must satisfy their employees’ needs to fulfill their filial obligations to enhance their job satisfaction and loyalty to the organization. In contemporary China, some employers have improved the job satisfaction and organizational commitment of their employees by regularly helping their parents travel and meet them and by organizing family reunions.

Taken together, the results of the present study serve as an empirical base upon which a sound and evidence-based retention management plan for employees who work in Chinese manufacturing enterprises can be founded.

## Figures and Tables

**Figure 1 ijerph-18-00714-f001:**
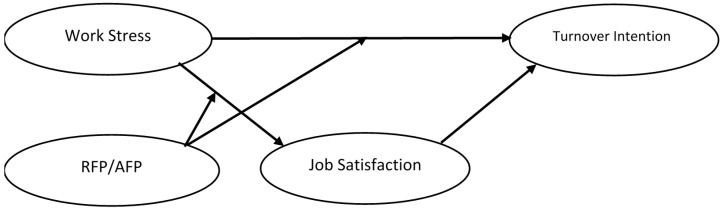
Theoretical model.

**Figure 2 ijerph-18-00714-f002:**
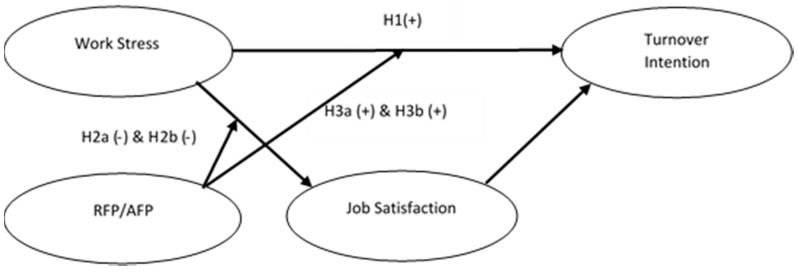
Empirical model.

**Table 1 ijerph-18-00714-t001:** Mean, standard deviations, scale reliabilities, and correlations matrix for the whole sample.

	*M*	*SD*	1	2	3	4	5	6	7	8	9	10
1. Age in years	28.06	6.21	_									
2. Gender	0.55	0.50	−0.04	_								
3. Education	0.54	0.50	0.11	0.04	_							
4. Tenure with present employer	31.03	42.47	0.53	0.02	0.30	_						
5. Marital status	1.56	0.52	−0.54	0.12	0.07	−0.30	_					
6. Reciprocal filial piety (RFP)	6.51	0.71	−0.04	−0.01	0.03	−0.02	0.05	(0.82)				
7. Authoritarian filial piety (AFP)	4.42	1.12	−0.09	0.24	−0.11	−0.01	0.05	0.16	(0.67)			
8. Work stress	4.08	1.31	−0.16	0.16	−0.09	−0.02	0.10	0.04	0.31	(0.87)		
9. Job satisfaction	4.81	1.34	0.17	0.05	0.08	0.13	−0.14	0.10	0.07	−0.30	(0.92)	
10. Turnover intention	3.72	1.39	−0.18	0.07	−0.25	−0.13	0.17	−0.09	0.16	0.37	−0.58	(0.93)

*N* = 506; Cronbach’s Alphas on the diagonal (“_” means not applicable); *r* > |0.086|, *p* < 0.05; *r* > |0.115|, *p* < 0.01; *r* > |0.158|, *p* < 0.001 (two–tailed); Gender (1 = male, 0 = female); Education (1 = lower education: primary or below, junior middle school and high school, 0 = higher education: junior college, undergraduate and graduate); Tenure with present employer (months); Marital status (1 = married, 2 = unmarried, 3 = divorced).

**Table 2 ijerph-18-00714-t002:** Hierarchical regression results for testing mediation.

	1 Turnover Intention				2 Job Satisfaction	
Variables and Statistic	Step 1	Step 2	Step 3	Step 4	Step 1	Step 2
Age	−0.10	−0.04	−0.04	−0.01	0.11	0.05
Gender	0.05	0.01	0.09 *	0.06	0.06	0.11 *
Education	−0.26 ***	−0.23 ***	−0.22 ***	−0.21 ***	0.07	0.04
Tenure	0.06	0.02	0.07	0.05	0.02	0.06
Marital status	0.16 **	0.15 **	0.11 *	0.11 *	−0.09	−0.08
Work stress		0.32 ***		0.18 ***		−0.30 ***
Job satisfaction			−0.57 ***	−0.52 ***		
*F*	11.58 ***	20.72 ***	57.11 ***	53.84 ***	4.35 ***	11.83 ***
*R*^2^ (Adj. *R*^2^)	0.11 (0.10)	0.21 (0.20)	0.42 (0.41)	0.44 (0.43)	0.04 (0.03)	0.13 (0.12)

*N* = 506, *** *p* < 0.001, ** *p* < 0.01, * *p* < 0.05; Control variables include age, gender, education, tenure, and marital status; significance testing of *R*^2^ is compared to the control model.

**Table 3 ijerph-18-00714-t003:** Regression results for testing moderation effect of reciprocal filial piety (RFP) and authoritarian filial piety (AFP) on the relationship between work stress and job satisfaction, and between work stress and turnover intention.

		Job Satisfaction			Turnover Intentions		
Step	Variables	β	*R*^2^ (Adj. *R*^2^)	F	β	*R*^2^ (Adj. *R*^2^)	F
1	Control variables		0.04 (0.03)	4.35 ***		0.11 (0.10)	11.58 ***
2	Work stress	−0.35 ***	0.17 (0.15)	11.73 ***	0.32 ***	0.22 (0.21)	16.72 ***
	RFP	0.10 *			−0.11 **		
	AFP	0.16 **			0.05		
3	Work stress X RFP	−0.19 ***	0.08 (0.07)	6.76 ***	0.15 **	0.13 (0.12)	11.90 ***
4	Work stress X AFP	−0.15 **	0.07 (0.05)	5.59 ***	0.18 **	0.14 (0.13)	13.06 ***

*N* = 506, *** *p* < 0.001, ** *p* < 0.01, * *p* < 0.05; Control variables include age, gender, education, tenure, and marital status; *R*^2^ significance testing for Models 3 and 4 is compared to Model 2.

**Table 4 ijerph-18-00714-t004:** Moderated mediation results for job satisfaction across levels of reciprocal filial piety (RFP) and authoritarian filial piety (AFP) on turnover intentions.

		Turnover Intentions			
Moderator	Level	Conditional Indirect Effect	SE	Z	*p*
RFP	High	0.26	0.08	3.29	<0.01
	Low	−0.01	0.02	−0.33	0.74
AFP	High	0.21	0.06	3.17	<0.01
	Low	0.06	0.07	0.82	0.41

*N* = 506, *** *p* < 0.001, ** *p* < 0.01, * *p* < 0.05.

## Data Availability

The data presented in this study are available on request from the corresponding author. The data are not publicly available due to privacy or ethical considerations.

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
