# Peer review of "The Role of Filial Piety in the Relationships between Work Stress, Job Satisfaction, and Turnover Intention: A Moderated Mediation Model"

_ijerph, 2021, doi:10.3390/ijerph18020714_

Round 1

Reviewer 1 Report

In my opinion, the paper would find a lot of readers. The study is original, interesting, and relevant in aspects of organization management and human resource management not only for China but also for other countries, despite cultural differences and differences in traditions.

It is necessary to correct language errors. Common typing and spelling mistakes in the work.

Author Response

Comments and Suggestions for Authors

In my opinion, the paper would find a lot of readers. The study is original, interesting, and relevant in aspects of organization management and human resource management not only for China but also for other countries, despite cultural differences and differences in traditions.

Dear Reviewer, thank you very much for your insightful comments and helpful suggestions, and your positive comments about our paper.

It is necessary to correct language errors. Common typing and spelling mistakes in the work.

According to your suggestions, we have asked another proofreader to go conscientiously through the whole text once more.

Reviewer 2 Report

Interesting paper. Authors should include all the items of the test they have used in an appendix. 

Author Response

Comments and Suggestions for Authors

Interesting paper. Authors should include all the items of the test they have used in an appendix.

Dear Reviewer, thank you very much for your insightful comments and valuable suggestions.

Concerning all the items of test, we have referred to the original sources of the scales carefully, and it is not possible due to copyright issues to take them all up in an Appendix.

Reviewer 3 Report

Thank you for the opportunity to review the manuscript entitled “The Role of Filial Piety in the Relationships between Work Stress, Job Satisfaction, and Turnover Intention: A Moderated Mediation Model” (ijerph-1022714).

The authors tested a moderated mediation model of the association between work stress with turnover intention by job satisfaction and filial piety in cross-sectional data from 506 Chinese employees. The data confirmed the hypotheses that job satisfaction serves as mediator and filial piety appears to be a moderator.

The manuscript is well drafted, analyzed and written and is interesting to the readership of the International Journal of Environmental Research and Public Health. I only have minor points for improvement.

In line 33 & 34, some words can not be read: “a :s ssence, ty,ltsstory: society-n” “humaity.onsho”, “text. parentsoblematic.filial”

Could you please provide more information on the acquaintances online sample, e.g., what was the response rate?

The authors might discuss in more detail why the two types of filial piety (reciprocal filial piety (RFP) and authoritarian filial piety (AFP)), which are only moderately related (r = .16, Table 1), yielding similar results and what the implications might be.

Author Response

Comments and Suggestions for Authors

Thank you for the opportunity to review the manuscript entitled “The Role of Filial Piety in the Relationships between Work Stress, Job Satisfaction, and Turnover Intention: A Moderated Mediation Model” (ijerph-1022714).

The authors tested a moderated mediation model of the association between work stress with turnover intention by job satisfaction and filial piety in cross-sectional data from 506 Chinese employees. The data confirmed the hypotheses that job satisfaction serves as mediator and filial piety appears to be a moderator.

The manuscript is well drafted, analyzed and written and is interesting to the readership of the International Journal of Environmental Research and Public Health. I only have minor points for improvement.

Dear Reviewer, thank you very much for your helpful comments and valuable suggestions.

In line 33 & 34, some words can not be read: “a :s ssence, ty,ltsstory: society-n” “humaity.onsho”, “text. parentsoblematic.filial”

Dear Reviewer, thank you very much for your notice. The random characters in line 33 & 34 have been eliminated and all has been adjusted carefully in line 34 & 35.

Could you please provide more information on the acquaintances online sample, e.g., what was the response rate?

Dear Reviewer, full anonymity was guaranteed but as our sample comprised people from our own network, our calls for participation have resulted into a maximum response rate.

The authors might discuss in more detail why the two types of filial piety (reciprocal filial piety (RFP) and authoritarian filial piety (AFP)), which are only moderately related (r = .16, Table 1), yielding similar results and what the implications might be.

Dear Reviewer, RFP and AFP are indeed moderately related in our research, which is consistent with Yeh and Bedford (2003) and Yeh et al.’s (2013) findings. The fact that RFP and AFP have yielded similar results in this empirical study, has been discussed in line 439 & 440 and line 448 & 449, namely, people with high levels of filial piety (i.e., RFP, AFP) may invest more time and energy in fulfilling parental demands, which in turn may affect job satisfaction and turnover intention.

Reviewer 4 Report

The present paper is an interesting investigation into the relationship between family obligations and working life in China. I found the paper well written and very topical

Some suggestions for improvement:

  1. There seems to be some sort of technical problems with the text in the introduction (rows 33-35). Please sort that out.
  2. Although the concept of filial piety is very well explained later in the text, i would put in a very short explanation of the concept in the abstract.
  3. The authors should consider whether H1 is really necessary. The point of the paper can perhaps be made with a less complicated model. We already know that work stress is negatively related to job satisfaction.
  4. I would like to see more about the representativeness of the sample, for instance in terms of age, education level and the like. Is it representative of industrial workers in China or the region? 
  5. An important point: why is there no question about the death of parents? Clearly, the situation for workers that have dead parents must be very different from those whose parents are alive?  Further, what about the distance in residence between workers and parents? If the parents live very far away, it is less possible to care for them etc? 

Author Response

Comments and Suggestions for Authors

The present paper is an interesting investigation into the relationship between family obligations and working life in China. I found the paper well written and very topical.

Dear Reviewer, thank you very much for your positive feedback and the time that has been given to our paper.

Some suggestions for improvement:

Dear Reviewer, thank you very much for your valuable suggestions.

1. There seems to be some sort of technical problems with the text in the introduction (rows 33-35). Please sort that out.

Dear Reviewer, thank you very much for your notice, we have canceled these random characters in line 33 & 34 and replaced carefully in line 34 & 35.

2. Although the concept of filial piety is very well explained later in the text, i would put in a very short explanation of the concept in the abstract.

Dear Reviewer, thank you very much for your suggestions, we have now put in a very short explanation of filial piety in the abstract (line 16).

3. The authors should consider whether H1 is really necessary. The point of the paper can perhaps be made with a less complicated model. We already know that work stress is negatively related to job satisfaction.

Dear Reviewer, thank you very much for your suggestions. We thought H1 is really necessary because we wanted to know whether the mediated effect would be moderated by RFP/AFP.

4. I would like to see more about the representativeness of the sample, for instance in terms of age, education level and the like. Is it representative of industrial workers in China or the region?

Dear Reviewer, thank you very much for your comments. Concerning the representativeness of the sample, we explicitly collected data from small and medium-sized enterprises that have survived more than ten years.

5. An important point: why is there no question about the death of parents? Clearly, the situation for workers that have dead parents must be very different from those whose parents are alive? Further, what about the distance in residence between workers and parents? If the parents live very far away, it is less possible to care for them etc?

Dear Reviewer, thank you very much for your comments, which we definitely understand and appreciate. However, in China, it is very impolite to ask whether one’s parents are alive. In Chinese traditional views of filial piety, there is a saying, “While his parents are alive, the son may not go abroad to a distance. If he has to go abroad, he must have a fixed place to which he goes”.